# The Differences in Clinical Presentation, Management, and Prognosis of Laboratory-Confirmed COVID-19 between Pregnant and Non-Pregnant Women: A Systematic Review and Meta-Analysis

**DOI:** 10.3390/ijerph18115613

**Published:** 2021-05-24

**Authors:** Durray Shahwar A. Khan, Areeba N. Pirzada, Anna Ali, Rehana A. Salam, Jai K. Das, Zohra S. Lassi

**Affiliations:** 1Department of Maternal and Child Health, Aga Khan University, Karachi 74800, Pakistan; durrayshahwar.a@gmail.com (D.S.A.K.); areeba.pirzada@scholar.aku.edu (A.N.P.); asalam.rehana@gmail.com (R.A.S.); jai.das@aku.edu (J.K.D.); 2Robinson Research Institute, University of Adelaide, Adelaide 5005, Australia; anna.ali@adelaide.edu.au

**Keywords:** COVID-19, SARS-CoV-2, coronavirus 2, pregnant, non-pregnant adults, child-bearing age women

## Abstract

Background: The coronavirus disease 2019 (COVID-19) pandemic has affected millions of people across the globe. Previous coronavirus outbreaks led to worsened symptoms amongst pregnant women, suggesting that pregnant women are at greater risk. Objectives: Our aim is to investigate the differences in clinical presentation, management, and prognosis of COVID-19 infection in pregnant and non-pregnant women. Methods: We ran a search on electronic databases and analysis of the relevant articles was done using Revie Manager 5.4. Results: The review consists of nine studies comprising 591,058 women (28,797 pregnant and 562,261 non-pregnant), with most of the data derived from two large studies. The risk of experiencing fever (RR: 0.74; 95% CI: 0.64–0.85), headache (RR: 0.77; 95% CI: 0.74–0.79), myalgia (RR: 0.92; 95% CI: 0.89–0.95), diarrhea (RR: 0.40, 95% CI: 0.39–0.43), chest tightness (RR: 0.86; 95% CI: 0.77–0.95), and expectoration (RR: 0.45; 95% CI: 0.21–0.97) were greater amongst non-pregnant COVID-19-infected women. Pregnant women with COVID-19 were less likely to be obese (RR: 0.68; 95% CI: 0.63–0.73) or have a smoking history (RR: 0.32; 95% CI: 0.26–0.39). COVID-19-infected non-pregnant women had a higher frequency of comorbidity such as chronic cardiac disease (RR: 0.58; 95% CI: 0.44–0.77), renal disease (RR: 0.45; 95% CI: 0.29–0.71), and malignancy (RR: 0.82; 95% CI: 0.68–0.98), compared to COVID-19-infected pregnant women. The risk of ICU admission (RR: 2.26; 95% CI: 1.68–3.05) and requirement of invasive mechanical ventilation (RR: 2.68; 95% CI: 2.07–3.47) were significantly higher amongst pregnant women. Conclusions: Although the frequency of risk factors and the risk of experiencing clinical symptoms of COVID-19 were higher among non-pregnant women, COVID-19-infected pregnant women had a higher requirement of ICU admission and invasive mechanical ventilation compared to non-pregnant COVID-19-infected women. More well-conducted studies from varying contexts are needed to draw conclusions. Prospero registration: CRD42020204638.

## 1. Introduction

In December 2019, the coronavirus disease 2019 (COVID-19) first emerged as a cluster of pneumonia cases of unknown origin in Wuhan, China [1]. The disease, caused by the severe acute respiratory syndrome coronavirus 2 (SARS-CoV-2), was soon declared a “public health emergency of international concern” and later characterized as a pandemic by the World Health Organization (WHO) [2]. As of 13 October 2020, there have been a total of more than 40.2 million confirmed cases of COVID-19, with over 1.1 million deaths worldwide [3]. Since the global outbreak, several studies have been published reporting clinical characteristics, laboratory findings, and management associated with COVID-19 in the general population, focusing mainly on non-pregnant adults [4,5,6,7].

Pregnancy is a unique state in which the maternal immune system has to overcome two main challenges: protecting the fetus against an immunological attack while maintaining adequate defense against various microbial threats. Physiological and mechanical changes associated with gestation predispose pregnant women to severe forms of respiratory infections with subsequent higher maternal and fetal mortality [8,9]. During the last two decades, coronavirus has been responsible for two major epidemics; the severe acute respiratory syndrome (SARS-CoV) and Middle East respiratory syndrome (MERS-CoV), with a case fatality of 10.5% and 34.4%, respectively [10]. It was found that these infections were associated with worsening symptoms and clinical outcomes among pregnant women ranging from severe maternal illness to spontaneous abortion, and even maternal death [8,9,11] Although SARS-CoV-2 appears to be less virulent than the aforementioned coronaviruses, its spread is far more rapid and efficient among close contacts [12]. Therefore, it has raised additional concerns in pregnant women because previous experiences with both SARS-CoV and MERS-CoV have shown severe complications in this vulnerable population.

The increased risk of viral pneumonia in the obstetric population makes it imperative to evaluate whether there is any difference in the clinical course and outcomes between pregnant and non-pregnant women infected with COVID-19. Furthermore, we found no systematic review that provides a comparison of available evidence on COVID-19 among women of reproductive age based on their pregnancy status. Therefore, in this systematic review and meta-analysis, we aim to describe the clinical characteristics, management, and prognosis of COVID-19-infected pregnant women compared to COVID-19-infected non-pregnant women. In the next section, we specify the method with which this systematic review and meta-analysis was conducted, after which we have the results section, the discussion, and finally, the conclusion. The findings of this review will facilitate healthcare workers in understanding the disease, aid in the clinical management and counseling of these patients, as well as allow policy makers to form guidelines for the general public.

## 2. Methodology

This systematic review has been registered in the International Prospective Register of Systematic Reviews (PROSPERO) database with ID number CRD42020204638 and follows the recommendations established by the Preferred Reporting Items for Systematic Reviews and Meta-Analyses (PRISMA) [13] (Appendix A).

A systematic literature search was conducted until 25 February 2021, using PubMed, Embase, the WHO COVID-19 database, and Google Scholar. Furthermore, medRxiv and bioRxiv were screened for pre-print papers. The following terms and their variants were combined and used in devising the search strategy: “Pregnant women” OR “Pregnancy” AND “Coronavirus” OR “Covid-19” OR “SARS-CoV-2”. The full search strategy and terms used are available in Appendix A. We did not apply any language restrictions; however, papers published since 31 December 2019, were included.

All observational studies (cohort, case-control, cross-sectional, or case-series) including consecutive patients with a comparison group of pregnant and non-pregnant women of reproductive age group with laboratory-confirmed SARS-CoV-2 infection and reporting clinical characteristics, management, and prognosis were considered eligible. A case-series was defined as a study with a sample size of less than ten participants. Only studies that compared pregnant women with COVID-19 with non-pregnant women with COVID-19 were eligible for inclusion. We excluded studies describing only either pregnant or non-pregnant women with COVID-19. Studies were checked for data overlapping by assessing their center of data collection and the time period during which the data were collected. When it was unclear, authors were contacted to ensure that they reported results from different centers or during different time periods. Identified overlapping papers were further assessed and the studies with inclusion of more variables, bigger sample size, and better quality of assessment were chosen, as shown in Appendix A.

Two reviewers (D.S.A.K. and A.N.P.) independently screened the titles and abstracts for eligibility. After the initial search, full texts of relevant articles were examined for inclusion and exclusion criteria. Primary studies that fulfilled the inclusion criteria were selected for this systematic review. Any disagreement among the authors was resolved through consensus or consulting a senior reviewer (Z.S.L.).

Two authors (D.S.A.K. and A.N.P.) extracted relevant information independently from included studies. The following items were extracted from each study if available: author’s name, study design, country, duration of the study, setting, total number of study participants, demographics, past medical history, presenting signs and symptoms, management, and complications. For clinical presentation, data on the number of asymptomatic and COVID-19-like symptoms were extracted. Data on management with antivirals, antibiotics, corticosteroids, or any other new medication/technique were also recorded. Complications in the two groups, be it progression to severe COVID-19 infection, requirement for ventilation, intensive care unit (ICU) admission, or death, were also considered. The quality of the studies included in this meta-analysis was assessed using the National Heart, Lung, and Blood Institute (NHLBI) quality assessment tool for observational cohort and case-control studies [14]. This tool helps evaluate the internal validity of a study, hence ensuring that the results are truly due to the exposure being evaluated.

Data were entered and analyzed using Review Manager (RevMan) version 5.4 [15]. Mean difference (MD) with 95% confidence intervals (CI) was used for continuous data and relative risk (RR) with 95% CI for dichotomous data. Random effect models were used and heterogeneity between the studies was explored using the *p*-value of Chi^2^ and I^2^. Sensitivity analysis was performed by removing one large study.

## 3. Results

There was a total of 4347 titles identified after the initial search, 4179 articles were excluded after screening titles and abstracts. Of the 168 studies retrieved for full text review, only 9 were found eligible for inclusion [16,17,18,19,20,21,22,23,24]. We excluded 159 studies on full-text review, of which 42 were authors’ perspectives or reviews, 44 were guidelines or guidance papers based on other coronavirus strains, 43 studies compared COVID-19 infected pregnant women with non-infected individuals or asymptomatic pregnant women, 26 studies did not have any outcomes of interest reported, and 4 were overlap studies conducted at the same center during the same time period, as shown in Figure 1.

### 3.1. Description of Included Studies

All nine studies included were observational studies, with six retrospective cohorts [16,17,18,19,20,21], one prospective cohort [24], and two case-control studies [22,23]. The data in the included studies were collected between December 2019 to February 2021 with eight papers published in the year 2020 and one in 2021 [24]. Five studies were conducted in China [16,17,18,20,22], one in Israel [21], one in Mexico [24], and two in the United States (US) [19,23]. Six studies were from a single center [16,17,18,20,21,22] whereas three were multicenter studies [19,23,24]. On methodological quality, it was identified that all studies mentioned their main objective, study population with uniform application of inclusion and exclusion criteria, consistency in their method of measuring exposure, and appropriate discussion of outcomes. However, only three studies took confounders into account [16,19,23].

The number of enrolled individuals in each study ranged from 36 to 409,462. Six studies had a sample size of less than 150 participants [16,17,18,20,21,22,23], whereas one study conducted in Mexico had a sample size of 181,088 [24] and another study conducted in the US had a total sample size of 409,462 [19], leading to the total number of participants included in this review to be 591,058. All the participants were COVID-19 positive with 28,797 pregnant women and 562,261 non-pregnant women. Characteristics of included studies are reported in Table 1 and their methodological quality in Table 2a,b. Results from the pooled analysis are presented in Table 3.

The most common method of confirming COVID-19 infection was via reverse transcriptase-polymerase chain reaction test (RT-PCR) of a swab sample from either the nasopharynx or the oropharynx. Five studies confirmed COVID-19 infection only through RT-PCR alone [18,21,22,23,24]. Two studies used both RT-PCR and serological markers (IgM and IgG antibodies) to confirm COVID-19 infection [16,20]. One study with a sample size of 64 reported testing via nucleic amplification and confirmed the testing with a chest computed tomography (CT) scan [17], whereas one study vaguely mentioned using molecular amplification detection test on clinical specimens of 409,462 individuals but did not specify details [19].

### 3.2. Findings

Our meta-analysis found that pregnant women with COVID-19 were 2.8 years younger compared to non-pregnant counterparts with COVID-19 (Table 3). Non-pregnant women were more commonly reported to be obese (RR 0.68; 95% CI 0.63 to 0.73) and have a smoking history (RR 0.32; 95% CI 0.26 to 0.39) compared to pregnant women with COVID-19 infection. Chronic cardiac disease (RR 0.58; 95% CI 0.44 to 0.77), renal disease (RR 0.45; 95% CI 0.29 to 0.71), and malignancy (RR 0.82; 95% CI 0.68 to 0.98) were more commonly present in COVID-19-infected non-pregnant women compared to pregnant women with COVID-19 infection. There was no difference in other reported co-morbidities including diabetes mellitus, chronic respiratory disease, hypothyroidism, mental sickness, and chronic hepatitis (Figure 2).

Overall, the most common symptoms were headache (four studies, 24.5%), cough (six studies, 23.1%), myalgia (four studies, 17.7%), and fever (seven studies, 17.5%). Pregnant women were at a lower risk of experiencing fever (RR 0.74; 95% CI 0.64 to 0.85; seven studies, 409,838 participants), headache (RR 0.77; 95% CI 0.74 to 0.79; four studies, 409,680 participants), myalgia (RR 0.92; 95% CI 0.89 to 0.95; four studies, 409,680 participants), diarrhea (RR 0.40, 95% CI 0.39 to 0.43; three studies, 409,569 participants), chest tightness (RR 0.86; 95% CI 0.77 to 0.95; three studies, 409,569 participants), and expectoration (RR 0.45; 95% CI 0.21 to 0.97; two studies, 107 participants) as compared to non-pregnant women. The risk of being asymptomatic (RR 3.94; 95% CI 1.69 to 9.20; three studies, 218 participants) was higher amongst pregnant women as compared to non-pregnant women. The risk of other symptoms such as cough, rhinorrhea, chills, fatigue, nausea and vomiting, rash, abdominal pain, dizziness, sore throat, shortness of breath, nasal congestion, and loss of appetite were similar across both groups (Figure 3).

The risk of ICU admission was found to be significantly higher amongst pregnant women (RR 2.26; 95% CI 1.68 to 3.05; five studies, 424,587 participants) and they were also more likely to receive invasive mechanical ventilation (RR 2.68; 95% CI 2.07 to 3.47; three studies, 409,616). However, no difference in risk was found in the severity of COVID-19 infection amongst pregnant and non-pregnant women. Severe COVID-19 was reported by two articles, one of which by Qiancheng et al. who defined it as “shortness of breath with a respiratory rate greater than 30 breaths/minute, or oxygen saturation less than 93% at rest, or alveolar oxygen partial pressure/faction of inspiration O_2_ (PaO_2_/FiO_2_) less than 300 mmHg” [16] (R Wang et al. failed to define the “severe” COVID-19 infection). The risk of maternal mortality (RR 1.08; 95% CI 0.89 to 1.31) was found to be equal amongst pregnant and non-pregnant women [18] (Figure 4).

Both groups were at an equal risk to be managed with oxygen therapy, antivirals, antibiotics, and Chinese medicine. However, the use of immunoglobulins (RR 0.46; 95% CI 0.26 to 0.81) was found to be lesser amongst pregnant females, whereas for corticosteroids (RR 1.61; 95% CI 1.02 to 2.55), it was higher amongst pregnant women as compared to their non-pregnant counterparts as shown in Figure 4.

#### Sensitivity Analysis

The studies conducted by Zambrano (2020) and Martinez-Portilla (2021) were removed and a sensitivity analysis was performed [19,24]. This was because a large sample size (n= 409,462 and n = 181,088, respectively) came from these studies, conducted across 50 states in the US and Mexico alone.

After removing Zambrano et al. and Martinez-Portilla et al., the number of enrolled individuals in each study ranged from 36 to 132. All the participants were COVID-19 positive with 180 pregnant women and 328 non-pregnant women, a combined total of 508 participants.

The findings were similar and suggested that pregnant women were at a lower risk of experiencing fever (RR 0.66; 95% CI 0.53 to 0.83; six studies, 376 participants), shortness of breath (RR 0.57; 95% CI 0.33 to 0.96; five studies, 340 participants), expectoration (RR 0.45; 95% CI 0.21 to 0.97; two studies, 107 participants), and chills (RR 0.03; 95% CI 0.00 to 0.52; one study, 43 participants) compared to non-pregnant women. The chances of being asymptomatic (RR 3.94; 95% CI 1.69 to 9.20, 3 studies, 218 participants) was higher amongst COVID-19-infected pregnant women compared to COVID-19-infected non-pregnant women. The risk of requiring ICU admission or management with medications was equal for both groups. However, the use of immunoglobulins among non-pregnant women was still higher as compared to pregnant women, while the use of corticosteroids was higher among pregnant women (Table 4).

## 4. Discussion

Human coronaviruses are among the most common pathogens causing viral respiratory infections. In the past two decades, the world has experienced three coronaviruses outbreaks, and the most recent strain, SARS-CoV-2, has led to the greatest public health crisis of the century. Amid this pandemic, the increasing mortality rate has called for a better understanding and protection of the vulnerable populations infected with the disease.

This systematic review summarizes the findings of 591,058 women with laboratory-confirmed COVID-19 infection, with 28,797 of them being pregnant. In the present meta-analysis; we found that in comparison with pregnant women, non-pregnant women are at a higher risk of experiencing symptoms such as headache, fever, expectoration, myalgia, chest tightness, wheezing, diarrhea, and anosmia, as primary symptoms of COVID-19. Non-pregnant women of reproductive age with COVID-19 had a higher frequency of comorbidities such as chronic cardiac diseases, renal diseases, and malignancy compared to pregnant COVID-19-infected women. The treatment modalities used in pregnant women were similar to the ones used in non-pregnant women, with a greater preference for corticosteroids in pregnant women. Pregnant women were more likely to be admitted to ICU and receive mechanical ventilation though there was no difference in the severity of the disease between both groups.

Pregnant women, due to their immunocompromised state, are more likely to experience complications of infectious diseases such as influenza, SARS, and MERS [17,18]. During the influenza A subtype H1N1 pandemic in 2009, pregnant women accounted for 5% of all H1N1-related deaths and were at an increased risk for severe disease, including hospitalization, ICU admissions, and death compared to their non-pregnant counterparts [11,25]. Similar trends were observed during the SARS and MERS outbreaks [9,26]. Lam, Chui Miu et al. [27] reported that 40% of the pregnant women affected with SARS required mechanical ventilation and had a case fatality of 30%, compared to 13% and 11% in non-pregnant individuals. Our review revealed a higher risk of ICU admissions in pregnant women; however, it did not show worsening clinical symptoms in pregnant women compared to non-pregnant women infected with COVID-19. Through the review, we found non-pregnant women to be at a higher risk of experiencing symptoms like headache, myalgia, fever, expectoration, chest tightness, wheezing, diarrhea, and anosmia compared to their pregnant counterparts. Many other studies and systematic reviews, however, reported clinical characteristics to be similar amongst COVID-19 infected pregnant and non-pregnant women [28,29,30,31,32]. Furthermore, in our review, both groups received similar supportive treatments irrespective of their pregnancy status. In concert with other studies [29,33,34,35], most patients received oxygen therapy in addition to antiviral and antibiotic medications. Our study demonstrated a greater likelihood of corticosteroid use in pregnant women. However, despite being the most commonly reported medication in another review too [36], the use of corticosteroids for COVID-19 has generally not been recommended in pregnant women, due to the increased risk of preterm birth, low birth weight, and pre-eclampsia associated with its use in pregnancy [37,38].

Some of the limitations identified for this review are (1) a small number of studies included, (2) smaller sample size for most studies with a smaller number of pregnant women compared to non-pregnant women, (3) two isolated studies with a large sample size having a greater impact on the overall result, (4) lack of data on other significant variables such as socioeconomic status and ethnicity, (5) studies from limited developed countries, and (6) unadjusted analysis in most studies. We also identified that apart from two studies, all the other studies had small sample sizes of less than 150. There is a need for studies with a bigger sample size and a comparable number of pregnant and non-pregnant COVID-19-infected women with adjusted analysis to reach more conclusive results for the future updates of this review. Moreover, there is a need to compare data on other variables, especially demographic variables such as socioeconomic status and ethnicity. The current manuscript reports data on ethnicity from a single study and therefore, is biased towards the data from the largest sample-sized study, leading to a lack of generalizability. Future studies in the domain should also highlight whether the ICU admission or worsening state in pregnant women was more likely to be due to COVID-19 infection or a delivery complication. Multivariable analysis to identify factors associated with clinical presentation, management, and prognosis in pregnant and non-pregnant women could not be done due to insufficient data. However, if data on individual patients are provided in the future, then individual patient data meta-analysis (IPD-MA) would be the ideal approach to providing insights into recognizing and managing COVID-19 infection in pregnant women. More studies on populations across the world need to be published to prevent the chance of bias towards a particular set of people. A future update of this systematic review may then be warranted and can, therefore, help reach conclusive findings.

## 5. Conclusions

In conclusion, the findings of this study summarize the epidemiological and clinical characteristics, along with the management and prognosis of women of reproductive age with COVID-19 based on their pregnancy status. With the disease burden increasing every day, these data equip healthcare workers to better identify and monitor the patients who are more susceptible to the disease and to make informed decisions when treating the patients.

## Figures and Tables

**Figure 1 ijerph-18-05613-f001:**
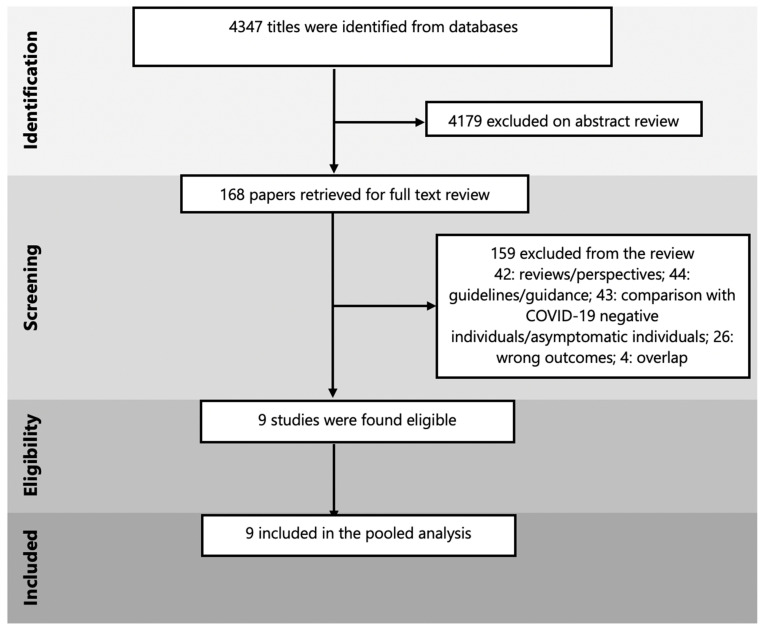
PRISMA flow diagram.

**Figure 2 ijerph-18-05613-f002:**
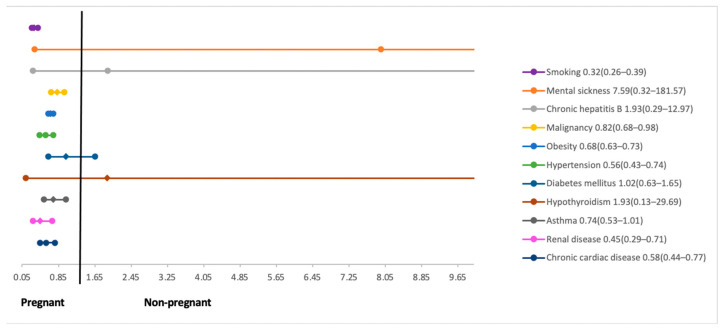
Past medical history among pregnant and non-pregnant women with COVID-19.

**Figure 3 ijerph-18-05613-f003:**
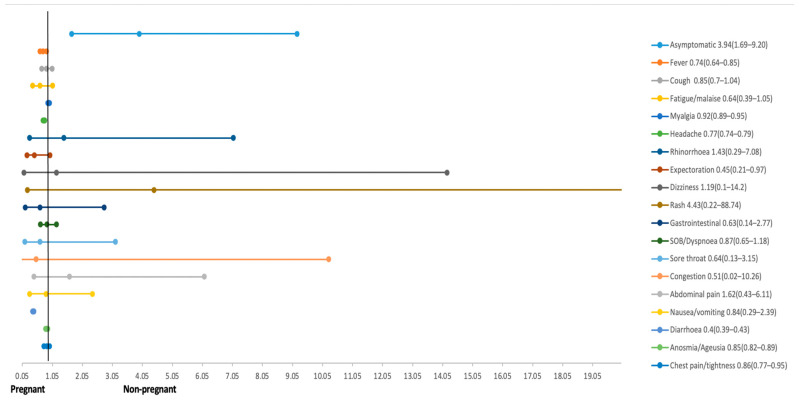
Clinical presentation among pregnant and non-pregnant women with COVID-19.

**Figure 4 ijerph-18-05613-f004:**
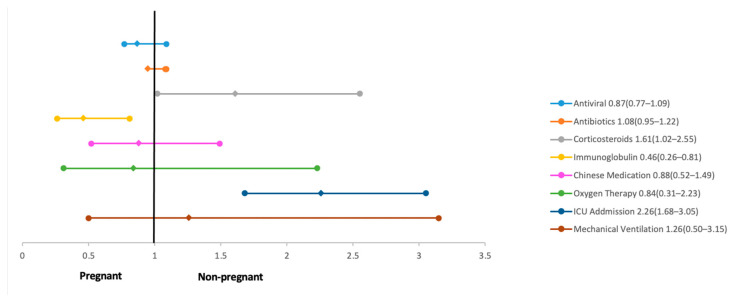
Management among pregnant and non-pregnant women with COVID-19.

**Table 1 ijerph-18-05613-t001:** Characteristics of included studies.

Study and Year	Study Design	Country and Time Period	Setting	Total Number	Demographics		Past Medical History		Presenting Signs and Symptoms		Management		Complications	
					Pregnant	Non-Pregnant	Pregnant	Non-Pregnant	Pregnant	Non-Pregnant	Pregnant	Non-Pregnant	Pregnant	Non-Pregnant
Xu Qiancheng 2020 [16]	Retrospective cohort	Wuhan, China 15 January to 15 March 2020	The Central Hospital of Wuhan	Total: 82 Pregnant: 28 Non-pregnant: 54	Mean age(years): 30 ± 1.28 Mean gestational age (weeks): 38 ± 0.61 First trimester: 3 Second trimester: 1 Third trimester: 24	Mean age (years): 31	Gestational hypertension: 1 Gestational diabetes: 2 Chronic hepatitis B: 2 Hypothyroid: 1	Hypertension: 0 Diabetes: 4 Chronic hepatitis B: 2 Hypothyroid: 1	Fever: 5 Malaise: 1 Cough: 7 Dyspnea: 2 Abdominal pain: 5	Fever: 29 Malaise: 3 Cough: 32 Dyspnea: 6 Abdominal pain: 0	Antiviral: 21 (Ribavirin: 20 Umifenovir: 1) Antibiotics: 24 (Cephalosporin: 20, Quinolone:4) Corticosteroid:4 Immunoglobulin: 3 Hospitalization: 7	Antiviral: 54 (Ribavirin: 19 Umifenovir: 11 Riba + Umi: 17 Triple combo with Interferon: 7) Antibiotics: 47 (Cephalosporin: 9, Quinolone: 6, Cephalosporin + Quinolone: 32) Corticosteroid: 21 Immunoglobulin: 19 Hospitalization: 0	Preterm Birth: 1	N/A
Shuang Xu 2020 [17]	Retrospective Cohort	Wuhan, China 15 January to 15 March 2020	Union Hospital	Total: 64 Pregnant: 34 Non-pregnant: 30	Mean age (years): 30 ± 4.26 First trimester and second: 8 Third trimester: 26 Exposure history: 10	Mean age: 34.77 ± 3.71 Exposure history: 2	GDM: 2 Hypothyroidism: 1 Pre-eclampsia: 1 Fetal distress: 1 Hypertension: 0 Cardiovascular: 1 Diabetes: 2 PROM: 4 Scarred uterus: 9	Hypertension: 0 Diabetes: 0 Cardiovascular: 0	Asymptomatic: 5 Fever: 22 Cough: 22 Fatigue: 6 Sputum: 5 SOB: 7 Chest tightness: 3 Headache: 5 Myalgia: 3 Nausea/vomiting: 2 Abdominal pain: 3 Diarrhea: 2 Rash: 2	Asymptomatic: 0 Fever: 26 Cough: 23 Fatigue: 16 Sputum: 13 SOB: 10 Chest tightness: 6 Headache: 7 Myalgia: 4 Nausea/vomiting: 5 Abdominal pain: 2 Diarrhea: 3 Rash: 0	Antibiotic: 30 Antiviral: 17 Corticosteroid: 19 Chinese medicine: 15 Oxygen therapy: 15 ICU admission: 1	Antibiotic: 24 Antiviral: 21 Corticosteroid: 9 Chinese medicine: 15 Oxygen therapy: 12 ICU admission: 0	Scarred uterus: 9 Gestational Diabetes: 2 Preeclampsia: 1 ICU admission: 1 Preterm Birth: 5 Post-partum fever: 3 NO neonatal complications	N/A
Shaoshuai Wang 2020 [18]	Retrospective Cohort	Wuhan, China 19 January to 2 March 2020	Tongji Hospital, Tongji Medical College of Huazhong University of Science and Technology	Total: 43 Pregnant: 17 Non-pregnant: 26	Mean age (years): 33.0 First trimester: 1 Second trimester: 3 Third trimester: 13 Healthcare workers: 3	Mean age (years): 33.5 Health care workers: 5	N/A	N/A	Fever: 8 Chills and rigors: 0 Headache: 0 Dizziness: 1 Fatigue: 1 Cough: 9 Expectoration: 3 Chest tightness: 2 SOB: 1 Myalgia: 0 Diarrhea: 1 Asymptomatic: 2 Abdominal pain: 4 Vaginal bleeding: 1 Reduced fetal movement: 1 Increased fetal movement: 1	Fever: 18 Chills and rigors: 2 Headache: 1 Dizziness: 0 Fatigue: 4 Cough: 12 Expectoration: 6 Chest tightness: 3 SOB: 1 Myalgia: 1 Diarrhea: 4 Asymptomatic: 2	Antiviral therapy: 13 Antibiotic therapy: 13 Glucocorticoid therapy: 4 Immunoglobulin therapy: 1 Cough suppressant therapy: 6 Oxygen support (nasal cannula): 6 Mechanical ventilation: 0	Antiviral therapy: 25 Antibiotic therapy: 23 Glucocorticoid therapy: 5 Immunoglobulin therapy: 3 Cough suppressant therapy: 18 Oxygen support (nasal cannula): 14 Mechanical ventilation: 0	Preterm birth: 2 ICU admission: 0 Death: 0	ICU Admission: 0 Death: 0
Laura Zambrano 2020 [19]	Retrospective Cohort	Unites States of America, 22 January to 3 October 2020	CDC directory via National Notifiable Diseases Surveillance System	Total: 409,462 Pregnant: 23,434 Non-pregnant: 386,028	Race/Ethnicity (%) Hispanic or Latino: 29.7 Asian: 2.4 Black: 14.5 White: 23.5 Multiple or another race: 3.1	Race/Ethnicity (%) Hispanic or Latino: 22.2 Asian: 2.2 Black: 14 White: 32.2 Multiple or another race: 3.2	Known underlying medical condition status: 7795 Diabetes mellitus: 427 Chronic lung disease: 506 Cardiovascular: 304 Chronic renal disease: 18 Chronic liver disease: 17 Immunocompromised: 124 Psychiatric disorder: 62 Autoimmune disorder: 26 Severe obesity: 174	Known underlying medical condition status: 160,065 Diabetes mellitus: 6119 Chronic lung disease: 9185 Cardiovascular: 7703 Chronic renal disease: 680 Chronic liver disease: 350 Immunocompromised: 2496 Psychiatric disorder: 1139 Other chronic disease: 1586 Autoimmune disorder: 515 Severe obesity: 1810	Cough: 5230 Fever: 3328 Muscle aches: 3818 Chills: 2537 Headache: 4447 SOB: 2692 Sore throat: 2955 Diarrhea: 1479 Nausea/vomiting: 2052 Abdominal pain: 870 Runny nose: 1328 New loss of taste or smell: 2234 Fatigue: 1404 Wheeze: 172 Chest pain: 369	Cough: 89,422 Fever: 68,536 Muscle aches: 78,725 Chills: 50,836 Headache: 95,713 SOB: 43,234 Sore throat: 60,218 Diarrhea: 38,165 Nausea/vomiting: 28,999 Abdominal pain: 16,123 Runny nose: 22,750 New loss of taste or smell: 43,256 Fatigue: 29,788 Wheeze: 3743 Chest pain: 7079	N/A	N/A	ICU admissions: 245 Mechanical Ventilation: 67 Death: 34	ICU admissions: 1492 Mechanical Ventilation: 412 Death: 447
Biheng Cheng 2020 [20]	Retrospective Cohort	Wuhan, China 15 January to 23 February 2020	Renmin Hospital of Wuhan University	Total: 111 Pregnant: 31 Non-pregnant: 80	Median age (years): 29.0 First trimester: 5 Second trimester: 6 Third trimester: 20	Median age (years) 33.0	Cardiovascular disease: 1 Respiratory disease: 0 Diabetes: 3 Malignancy: 0 Renal disease: 1 Gastric ulcer: 0 Mental sickness: 1	Cardiovascular disease: 4 Respiratory disease: 1 Diabetes: 1 Malignancy: 1 Renal disease: 1 Gastric ulcer: 1 Mental sickness: 0	Fever: 15 Cough: 14 Nasal congestion: 0 Rhinorrhea: 1 Myalgia: 1 Sore throat:1 Headache: 0 Dizziness: 0 SOB: 5 Digestive tract symptoms 3 Asymptomatic: 9 Asthenia: 1	Fever: 49 Cough: 48 Nasal congestion: 2 Rhinorrhea: 0 Myalgia: 8 Sore throat: 13 Headache: 2 Dizziness: 3 SOB: 30 Digestive tract symptoms 23 Asymptomatic: 5 Asthenia: 27	Antiviral: 29 Oseltamivir: 16 Arbidol: 25 Ribavirin: 8 IV Antibiotics: 29 Antifungal: 0 Corticosteroid: 20 Oxygen therapy: 2 Invasive ventilation: 0 Non-invasive ventilation: 0 ECMO: 0 Immunoglobulin: 7	Antiviral: 75 Oseltamivir: 24 Arbidol: 67 Ribavirin: 14 IV Antibiotics:60 Antifungal: 0 Corticosteroid:21 Oxygen therapy: 35 Invasive ventilation: 0 Non-invasive ventilation: 0 ECMO: 0 Immunoglobulin: 34	ICU admission: 0 Use of CRRT: 0	ICU admission: 1 Use of CRRT: 1
Aya Mohr-Sasson 2020 [21]	Retrospective Cohort	Israel, March to April 2020	Sheba Medical Center (University Affiliated Tertiary Medical Center)	Total: 36 11 pregnant 25 non-pregnant	Median Age: 28 All in third Trimester	Median age: 40	N/A	N/A	Fever: 3/11 Weakness: 5/11 Respiratory: 6/11 Gastrointestinal: 2/11 Others: 2/11	Fever: 15/25 Weakness: 16/25 Respiratory: 20/25 Gastrointestinal: 3/25 Others: 7/28	Hospitalization: 7 Home surveillance: 4	Hospitalization: 20 Home surveillance: 4	Intubation: 1 C-section: 2/11 (one due to symptoms related to COVID-19 and other due to non-reassuring fetal monitor)	Intubation: 1
Fang Liu 2020 [22]	Retrospective case-control study	Shanghai and Wuhan, China 23 January to 4 March 2020	Xinhua Hospital and Maternal and Child Health Hospital	Total: 40 Pregnant: 21 Non-pregnant: 19	Mean age: 31	Mean age: 31	N/A	N/A	Fever: 8/21 Cough: 6/21 SOB: 1/21 Fatigue: 8/21 Loss of appetite: 2/21	Fever: 14/19 Cough: 8/19 SOB: 1/19 Fatigue: 3/19 Loss of appetite: 0/19	N/A	N/A	ICU admission: 1/21 Mechanical ventilation: 1/21	ICU Admission: 1/19 Mechanical Ventilation: 1/19
Chelsea DeBolt 2020 [23]	Retrospective case-control study	New York and Philadelphia, United States 12 March to 5 May 2020	NYU Langone Health, Mount Sinai Hospital, Elmhurst Hospital, Montefiore Medical center, Thomas Jefferson University Hospital	Total: 132 Pregnant: 38 Non-pregnant: 94	Mean age: 34.7 Mean BMI: 31.7	Mean age: 37.9 Mean BMI: 33.4	N/A	N/A	N/A	N/A	Hydroxychloroquine: 34 Azithromycin: 25 Antivirals: 7 Tocilizumab: 3 Systemic steroids: 4 Convalescent plasma: 2 Therapeutic anticoagulation: 8 Prophylactic anticoagulation: 24	Hydroxychloroquine:76 Azithromycin: 56 Antivirals: 6 Tocilizumab: 4 Systemic steroids: 15 Convalescent plasma: 4 Therapeutic anticoagulation: 20 Prophylactic anticoagulation: 61	N/A	N/A
Martinez-Portilla2021 [24]	Prospective cohort study	Mexico, 1 February to 28 October 2020	Mexican National Registry of Coronavirus	Total: 181,088 Pregnant: 5183 Non-pregnant: 175905	Mean age: 28.5 ± 5.9	Mean age: 33.1 ± 7.5	COPD: 10 Asthma: 112 Smoker: 91 Hypertension: 150 Cardiovascular disease: 24 Obesity: 477 Diabetes: 174	COPD: 487 Asthma: Smoker: Hypertension: Cardiovascular disease: Obesity: Diabetes:	N/A	N/A	N/A	N/A	Death: 77 ICU admission: 154	Death: 2589 ICU admission: 941

**Table 2 ijerph-18-05613-t002:** (**a**) NHLBI quality assessment tool for cohort studies. (**b**) NHLBI quality assessment tool for case-control studies.

(**a**)							
**Study ID**	**Qiancheng 2020 [18]**	**Xu 2020 [19]**	**Wang 2020 [20]**	**Zambrano 2020 [21]**	**Cheng 2020 [22]**	**Mohr-Sasson 2020 [23]**	**Martinez-Portilla 2020 [24]**
1. Was the research question or objective in this paper clearly stated?	Yes	Yes	Yes	No	Yes	Yes	Yes
2. Was the study population clearly specified and defined?	Yes	Yes	Yes	Yes	Yes	Yes	Yes
3. Was the participation rate of eligible persons at least 50%?	N/A	N/A	N/A	N/A	N/A	N/A	N/A
4. Were all the subjects selected or recruited from the same or similar populations (including the same time period)? Were inclusion and exclusion criteria for being in the study prespecified and applied uniformly to all participants?	Yes	Yes	Yes	Yes	Yes	Yes	Yes
5. Was a sample size justification, power description, or variance and effect estimates provided?	No	No	No	No	No	No	No
6. For the analyses in this paper, were the exposure(s) of interest measured prior to the outcome(s) being measured?	No	No	No	No	No	No	No
7. Was the timeframe sufficient so that one could reasonably expect to see an association between exposure and outcome if it existed?	No	No	No	No	No	No	No
8. For exposures that can vary in amount or level, did the study examine different levels of the exposure as related to the outcome (e.g., categories of exposure, or exposure measured as continuous variable)?	N/A	N/A	N/A	N/A	N/A	N/A	N/A
9. Were the exposure measures (independent variables) clearly defined, valid, reliable, and implemented consistently across all study participants?	Yes	Yes	Yes	Yes	Yes	Yes	Yes
10. Was the exposure(s) assessed more than once over time?	N/A	N/A	N/A	N/A	N/A	N/A	N/A
11. Were the outcome measures (dependent variables) clearly defined, valid, reliable, and implemented consistently across all study participants?	Yes	Yes	Yes	Yes	Yes	Yes	Yes
12. Were the outcome assessors blinded to the exposure status of participants?	N/A	N/A	N/A	N/A	N/A	N/A	N/A
13. Was loss to follow-up after baseline 20% or less?	N/A	N/A	N/A	N/A	N/A	N/A	N/A
14. Were key potential confounding variables measured and adjusted statistically for their impact on the relationship between exposure(s) and outcome(s)?	Yes	No	No	Yes	No	No	No
(**b**)							
**Study ID**	**Fang Liu 2020 [24]**				**Chelsea DeBolt 2020 [23]**		
1. Was the research question or objective in this paper clearly stated?	Yes				Yes		
2. Was the study population clearly specified and defined?	Yes				Yes		
3. Did the authors include a sample size justification?	No				No		
4. Were controls selected or recruited from the same or similar population that gave rise to the cases (including the same timeframe)?	Yes				Yes		
5. Were the definitions, inclusion and exclusion criteria, algorithms or processes used to identify or select cases and controls valid, reliable, and implemented consistently across all study participants?	Yes				Yes		
6. Were the cases clearly defined and differentiated from controls?	Yes				Yes		
7. If less than 100 percent of eligible cases and/or controls were selected for the study, were the cases and/or controls randomly selected from those eligible?	Yes				Yes		
8. Was there use of concurrent controls?	No				No		
9. Were the investigators able to confirm that the exposure/risk occurred prior to the development of the condition or event that defined a participant as a case?	No				Yes		
10. Were the measures of exposure/risk clearly defined, valid, reliable, and implemented consistently (including the same time period) across all study participants?	Yes				Yes		
11. Were the assessors of exposure/risk blinded to the case or control status of participants?	No				No		
12. Were key potential confounding variables measured and adjusted statistically in the analyses? If matching was used, did the investigators account for matching during study analysis?	No				Yes		

**Table 3 ijerph-18-05613-t003:** Comparison of pregnant and non-pregnant women with COVID-19: summary estimates.

Outcomes	Relative Risk/Mean Difference (95% CI)	No of Studies; No of Participants	Heterogeneity Chi^2^ *p* Value; I^2^
**Demographics**			
Mean age (years)	−2.87 (−4.61 to −1.12)	6; 181,520	<0.00001; 92%
Advance maternal age (years)	0.55 (0.53 to 0.56)	1; 409,462	N/A
Mean BMI	−1.70 (−3.82 to 0.42)	1; 132	N/A
Obesity	0.68 (0.63 to 0.73)	2; 590,550	<0.00001; 99%
Smoking	0.32 (0.26 to 0.39)	2; 181,220	0.80; 0%
Hispanic/Latino	1.34 (1.31 to 1.37)	1; 409,462	N/A
Asian	1.07 (0.99 to 1.17)	1; 409,462	N/A
Black	1.03 (1.00 to 1.06)	1; 409,462	N/A
White	0.73 (0.71 to 0.75)	1; 409,462	N/A
Other (mix)	0.93 (0.87 to 1.00)	1; 409,462	N/A
**Clinical presentation**			
Asymptomatic	3.94 (1.69 to 9.20)	3; 218	0.47; 0%
Fever	0.74 (0.64 to 0.85)	7; 409,838	0.24; 24%
Cough	0.85 (0.70 to 1.04)	6; 409,802	0.13; 42%
Respiratory symptoms	0.68 (0.38 to 1.21)	1; 36	N/A
Rhinorrhea	1.43 (0.29 to 7.08)	2; 409,573	0.2; 39%
Expectoration	0.45 (0.21 to 0.97)	2; 107	0.30; 7%
Chills	0.22 (0.01 to 4.92)	2; 409,505	0.02; 81%
Headache	0.77 (0.74 to 0.79)	4; 409,680	0.96; 0%
Fatigue	0.64 (0.39 to 1.05)	7; 409,747	0.04; 54%
Myalgia	0.92 (0.89 to 0.95)	4; 409,680	0.71; 0%
Chest tightness	0.86 (0.77 to 0.95)	3; 409,569	0.59; 0%
Wheezing	0.76 (0.65 to 0.88)	1; 409,462	N/A
Diarrhea	0.40 (0.39 to 0.43)	3; 409,569	0.91; 0%
Nausea or vomiting	0.84 (0.29 to 2.39)	2; 409,526	0.13; 55%
Gastrointestinal	0.63 (0.14 to 2.77)	2; 147	0.13; 56%
Rash	4.43 (0.22 to 88.74)	1; 64	N/A
Dizziness	1.19 (0.10 to 14.20)	2; 154	0.25; 25%
Anosmia/ageusia	0.85 (0.82 to 0.89)	1; 409,462	N/A
Sore throat	0.64 (0.13 to 3.15)	2; 409,573	0.09; 66%
Shortness of breath	0.87 (0.65 to 1.18)	6; 409,802	0.32; 15%
Nasal congestion	0.51 (0.02 to 10.26)	1; 111	N/A
Abdominal pain	1.62 (0.43 to 6.11)	3; 409,608	0.09; 59%
Loss of appetite	4.55 (0.23 to 89.08)	1; 40	N/A
Other symptoms	0.65 (0.16 to 2.64)	1; 36	N/A
**Co-morbidities**			
Chronic cardiac disease	0.58 (0.44 to 0.77)	5; 590,807	0.02; 66%
Diabetes mellitus	1.02 (0.63 to 1.65)	5; 590,807	<0.00001; 90%
Chronic respiratory disease	0.74 (0.53 to 1.01)	4; 590,793	0.003; 79%
Renal disease	0.45 (0.29 to 0.71)	2; 409,573	0.21; 37%
Hypothyroidism	1.93 (0.13 to 29.69)	1; 82	N/A
Malignancy	0.82 (0.68 to 0.98)	2; 409,573	0.98; 0%
Mental sickness	7.59 (0.32 to 181.57)	1; 111	N/A
Chronic hepatitis B	1.93 (0.29 to 12.97)	1; 82	N/A
**Management**			
Oxygen therapy	0.84 (0.31 to 2.23)	4; 350	0.001; 81%
Antivirals	0.87 (0.70 to 1.09)	5; 432	0.009; 70%
Antibiotics	1.08 (0.95 to 1.22)	5; 432	0.17; 38%
Corticosteroids	1.61 (1.02 to 2.55)	5; 432	0.16; 39%
Immunoglobulin	0.46 (0.26 to 0.81)	3; 236	0.71; 0%
Chinese medicine	0.88 (0.52 to 1.49)	1; 64	N/A
**Complications**			
Severe COVID-19	1.60 (0.41 to 6.28)	2; 125	0.37; 0%
Maternal ICU admission	2.26 (1.68 to 3.05)	5; 424,587	0.02; 65%
Invasive ventilation	2.68 (2.07 to 3.47)	3; 409,616	N/A
Any ventilation	1.26 (0.50 to 3.15)	3; 15,082	0.03; 72%
Maternal death	1.08 (0.89 to 1.31)	2; 590,550	0.31; 4%

**Table 4 ijerph-18-05613-t004:** Summary estimates based on sensitivity analysis (removed Ellington 2020).

Outcomes	Relative Risk/Mean Difference (95% CI)	No. of Studies; No. of Participants	Heterogeneity Chi^2^ *p* Value; I^2^
**Demographics**			
Mean age (years)	−2.40 (−3.82 to −0.97)	5; 432	0.02; 67%
Mean BMI	−1.70 (−3.82 to 0.42)	1; 132	N/A
Smoking	0.25 (0.03 to 1.87)	1; 132	N/A
**Clinical presentation**			
Asymptomatic	3.94 (1.69 to 9.20)	3; 218	0.47; 0%
Fever	0.66 (0.53 to 0.83)	6; 376	0.32; 15%
Cough	0.77 (0.59 to 1.01)	5; 340	0.25; 26%
Respiratory symptoms	0.68 (0.38 to 1.21)	1; 36	N/A
Rhinorrhea	7.59 (0.32 to 181.57)	1; 111	N/A
Expectoration	0.45 (0.21 to 0.97)	2; 107	0.30; 7%
Chills	0.03 (0.00 to 0.52)	1; 43	N/A
Headache	0.60 (0.24 to 1.54)	3; 218	0.98; 0%
Fatigue	0.55 (0.25 to 1.24)	6; 376	0.03; 58%
Myalgia	0.52 (0.18 to 1.55)	3; 218	0.85; 0%
Chest tightness	0.60 (0.22 to 1.68)	2; 107	0.44; 0%
Diarrhea	0.49 (0.13 to 1.88)	2; 107	0.76; 0%
Nausea or vomiting	0.35 (0.07 to 1.69)	1; 64	N/A
Gastrointestinal	0.63 (0.14 to 2.77)	2; 147	2.25; 56%
Rash	4.43 (0.22 to 88.74)	1; 64	N/A
Dizziness	1.19 (0.10 to 14.20)	2; 154	0.25; 25%
Sore throat	0.20 (0.03 to 1.45)	1; 111	NA
Shortness of breath	0.57 (0.33 to 0.96)	5; 340	0.89; 0%
Nasal congestion	0.51 (0.02 to 10.26)	1; 111	N/A
Abdominal pain	4.20 (0.26 to 68.93)	2; 146	0.09; 65%
Loss of appetite	4.55 (0.23 to 89.08)	1; 40	N/A
Other symptoms	0.65 (0.16 to 2.64)	1; 36	N/A
**Co-morbidities**			
Chronic cardiac disease	1.53 (0.32 to 7.21)	3;257	0.50; 0%
Diabetes mellitus	2.45 (0.62 to 9.61)	3;257	0.30; 18%
Chronic respiratory disease	0.50 (0.19 to 1.29)	2; 243	N/A
Renal disease	2.58 (0.17 to 39.99)	1; 111	N/A
Hypothyroidism	1.93 (0.13 to 29.69)	1; 82	N/A
Malignancy	0.84 (0.04 to 20.17)	1; 111	N/A
Mental sickness	7.59 (0.32 to 181.57)	1; 111	N/A
Chronic hepatitis B	1.93 (0.29 to 12.97)	1; 82	N/A
**Management**			
Oxygen therapy	0.84 (0.31 to 2.23)	4; 350	0.001; 81%
Antivirals	0.87 (0.70 to 1.09)	5; 432	0.009; 70%
Antibiotics	1.08 (0.95 to 1.22)	5; 432	0.17; 38%
Corticosteroids	1.61 (1.02 to 2.55)	5; 432	0.16; 39%
Immunoglobulin	0.46 (0.26 to 0.81)	3; 236	0.71; 0%
Chinese medicine	0.88 (0.52 to 1.49)	1; 64	N/A
**Complications**			
Severe COVID-19	1.60 (0.41 to 6.28)	2; 125	0.37; 0%
Maternal ICU admission	1.83 (0.30 to 11.38)	3; 215	0.84; 0%
Any ventilation	2.28 (1.07 to 4.88)	2; 172	0.48; 0%

## Data Availability

The datasets used and/or analyzed during the current study are available from the corresponding author on reasonable request.

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
