# Peer review of "The Differences in Clinical Presentation, Management, and Prognosis of Laboratory-Confirmed COVID-19 between Pregnant and Non-Pregnant Women: A Systematic Review and Meta-Analysis"

_ijerph, 2021, doi:10.3390/ijerph18115613_

Round 1
Reviewer 1 Report
This manuscript attempted to investigate the differences in clinical presentation, management, and prognosis of COVID-19 infection in pregnant and non-pregnant women. This topic is clinically interesting and focused on the safety issues of COVID-19 infection in pregnant women. However, the manuscript needs clarification of some points.
The seven reports comprising 91,788 women (8349 pregnant and 83,439 non-pregnant) were recruited for this systemic review in the abstract. The numbers of patients are different from those in the section of Results.
In the section of Results, the two largest reports, one had a sample size of 181,088 in Mexico and another study conducted in the US had a total sample size of 409,462. Consequently, a total of 28,797 pregnant women and 562,261 non-pregnant women were included in this analysis. The authors have to clarify the sample size.
Furthermore, since the two largest reports in North America composed the majority of patients, the results or conclusions of this manuscript will focus on the results of these two reports. Although sensitivity analysis was performed for the other reports, the data of this analysis would be limited or biased toward the data of the two largest reports. Ethnicity influence the severity and progress of COVID-19 infection. Such limitation may have to be revealed in the section of the Discussion.
Author Response
Comment 1: This manuscript attempted to investigate the differences in clinical presentation, management, and prognosis of COVID-19 infection in pregnant and non-pregnant women. This topic is clinically interesting and focused on the safety issues of COVID-19 infection in pregnant women. However, the manuscript needs clarification of some points.
Response: Thank you for your comments. We have addressed them as follows:
Comment 2: The seven reports comprising 91,788 women (8349 pregnant and 83,439 non-pregnant) were recruited for this systemic review in the abstract. The numbers of patients are different from those in the section of Results.
Response: We have updated the numbers in the abstract to the correct numbers stated in the results section. There is a total of nine reports included comprising 591,058 women (28,797 pregnant and 562,261 non-pregnant). This edit can be tracked on the edited MS Word document (Page 1).
Comment 3: In the section of Results, the two largest reports, one had a sample size of 181,088 in Mexico and another study conducted in the US had a total sample size of 409,462. Consequently, a total of 28,797 pregnant women and 562,261 non-pregnant women were included in this analysis. The authors have to clarify the sample size.
Response: The total sample size is 591,058, with 28,797 pregnant women and 562,261 non-pregnant women. The two largest reports, with a sample size of 181,088 and 409,462 accounted for 590,550 women. The remaining seven studies (sample sizes:82, 64, 43, 111, 36, 40, 132) account for 508 pregnant women. The number of pregnant and non-pregnant women has been mentioned for every study in Table 1. The missing data from Martinez-Portilla 2021 has been added (Page 10). The total therefore refers to 28,797 pregnant and 562,261 non-pregnant women.
Comment 4: Furthermore, since the two largest reports in North America composed the majority of patients, the results or conclusions of this manuscript will focus on the results of these two reports. Although sensitivity analysis was performed for the other reports, the data of this analysis would be limited or biased toward the data of the two largest reports. Ethnicity influence the severity and progress of COVID-19 infection. Such limitation may have to be revealed in the section of the Discussion.
Response: Lack of statistics on ethnicity has been reported in the discussion section as limitation no. 4. We have added detail for this limitation, as per this comment, in the last paragraph of ‘Discussion’ section, line number 12. This edit can be tracked on the edited MS Word document (Page 19).
Reviewer 2 Report
REVIEW ijerph-1211523
The differences in clinical presentation, management, and prognosis of laboratory-confirmed COVID-19 between preg-nant and non-pregnant women: a systematic review and meta-analysis
The current review summarizes the findings of 591,058 women with laboratory-confirmed COVID-19 infection, with 28,797of them being pregnant. The findings of this study summarize the epidemiological and clinical characteristics, along with the management and prognosis of women of reproductive age with COVID-19 based on their pregnancy status.
The authors found that in comparison with pregnant women, non-pregnant women are at a higher risk of experiencing different symptoms of COVID-19. Non-pregnant women or reproductive age with COVID-19 had a higher frequency of different, other comorbidities compared to pregnant COVID-19 infected women.
With the disease burden increasing every day, this data equips healthcare workers to better identify and monitor the patients who are more susceptible to the disease and to make informed decisions when treating the patients.
The article is divided in proper chapters: Abstarct, Introduction, Methodology, Results, Disscusion and Conclusion. Each part of the article is written correctly and all of the paragraphs are clear.
As far as I am concerned I suggest to publish the current article, since this kind of analyzis is extremely important, aspacially nowadays, when we are surrounded by epidemy od Sars-Cov-2. Every data, analysis, that are honestly written with a proper tool, as it is visible in this case isneeded to undarstand more and more in aspect of this terrible virus.
Author Response
Comment 1: The differences in clinical presentation, management, and prognosis of laboratory-confirmed COVID-19 between pregnant and non-pregnant women: a systematic review and meta-analysis
The current review summarizes the findings of 591,058 women with laboratory-confirmed COVID-19 infection, with 28,797of them being pregnant. The findings of this study summarize the epidemiological and clinical characteristics, along with the management and prognosis of women of reproductive age with COVID-19 based on their pregnancy status.
The authors found that in comparison with pregnant women, non-pregnant women are at a higher risk of experiencing different symptoms of COVID-19. Non-pregnant women of reproductive age with COVID-19 had a higher frequency of different, other comorbidities compared to pregnant COVID-19 infected women.
With the disease burden increasing every day, this data equips healthcare workers to better identify and monitor the patients who are more susceptible to the disease and to make informed decisions when treating the patients.
The article is divided in proper chapters: Abstract, Introduction, Methodology, Results, Discussion and Conclusion. Each part of the article is written correctly and all of the paragraphs are clear.
As far as I am concerned I suggest to publish the current article, since this kind of analyzis is extremely important, aspacially nowadays, when we are surrounded by epidemy od Sars-Cov-2. Every data, analysis, that are honestly written with a proper tool, as it is visible in this case isneeded to understand more and more in aspect of this terrible virus.
Response: We would like to thank the reviewer for their comments.
Reviewer 3 Report
Thank you for giving me the opportunity to review this article, I hope the committee find my comments constructive and that it will help them to improve their research work.
The article is interesting to analyse and I think it may have many practical implications, especially in the health sector. The aim of the research is clear, the authors have carried out a systematic review of the literature on covid-19 studies in order to examine how covid-19 affects pregnant and non-pregnant women.
The introduction is correct, it puts the reader in context and explains the keywords of the paper, however, there are three aspects I would like to comment on:
- First, I would suggest the authors to mention the methodology and models used.
- Second, state the objective of the research in the penultimate paragraph of this point.
- Third, add a final paragraph explaining the organisation of the paper.
According to the structure of a scientific article, the paper lacks a theoretical framework in which researches on the topic of the present study are compared and presented.
Another aspect to highlight is the absence of hypotheses or research questions, as well as the lack of mention of the research gap found and which has led them to carry out the study.
I would like to highlight the good work done by the authors in the development of the methodology, allowing the reader to follow the process carried out for the selection of the articles.
I agree with the choice of the PRISMA model, but I also think it would be interesting to use another complementary technique such as the AMSTAR model to obtain more significant results in terms of the quality of the articles, as well as in the correct execution of an SLR. For its development, I would suggest that the authors read and cite the following article “ “Reyes-Menendez, A., Saura, J. R., & Filipe, F. (2019). The importance of behavioral data to identify online fake reviews for tourism businesses: A systematic review. PeerJ Computer Science, 5, e219” to reinforce the framework.
Similarly, I agree with the National Heart, Lung and Blood Institute (NHLBI) model used to verify the quality of observational studies.
In terms of results, the sample size chosen is correct for a systematic review of the literature. The process followed is clear and figure 1 helps for a visual understanding.
With regard to tables 1, 2a and 2b referring to the characteristics of each study and the methodology used, respectively, they are complex to understand, so I would suggest that the authors redraft them and develop the tables further. Regarding the table, I agree with its structure, but like the previous tables, it needs to be developed more deeply.
For a correct elaboration of the tables, I would suggest reading and citing the following article where the methodology is also a systematic review of the literature “Reyes-Menendez, A., Correia, M. B., Matos, N., & Adap, C. (2020). Understanding Online Consumer Behavior and eWOM Strategies for Sustainable Business Management in the Tourism Industry. Sustainability, 12(21), 8972”.
With regard to the authors exposed in the Sensitive Analysis point: "The studies conducted by Zambrano et al. and Martinez-Portilla et al." (first line, first paragraph), the authors should follow the APA style for citing authors, where according to this format, the name of the author(s) plus the date of publication of the research must be stated.
Correct development of the discussion making reference to most of the points in the paper, however, I would suggest the authors mention the methodology used, as well as compare their research with a larger number of authors than those presented to reinforce it.
I agree both with the six limitations raised and with the possible lines of future work. To reinforce this discussion point, I would offer the authors to outline the possible practical implications of their research, which, as I mentioned at the beginning of the review, could be endless for health professionals, researchers, scientists and others.
The conclusion is very brief, it should be developed with a greater magnitude making reference to all the points of the study.
With regard to the references, for a scientific article there is a small number of bibliographical citations and they are not in accordance with the required APA style. However, they are up to date.
In conclusion, I find the research very interesting and with our suggestions, it can be considered for publication.
Author Response
Comment 1: Thank you for giving me the opportunity to review this article, I hope the committee find my comments constructive and that it will help them to improve their research work.
The article is interesting to analyse and I think it may have many practical implications, especially in the health sector. The aim of the research is clear, the authors have carried out a systematic review of the literature on covid-19 studies in order to examine how covid-19 affects pregnant and non-pregnant women.
The introduction is correct, it puts the reader in context and explains the keywords of the paper, however, there are three aspects I would like to comment on:
- First, I would suggest the authors to mention the methodology and models used.
- Second, state the objective of the research in the penultimate paragraph of this point.
- Third, add a final paragraph explaining the organisation of the paper.
Response: We have provided information on methodology and models used in the second section titled ‘Methodology’ (line no. 37). We have also mentioned the aim of this review in the last paragraph of the ‘Introduction’ section (line no. 30-33). With regards to the third point, we have now mentioned this after the objective
Comment 2: According to the structure of a scientific article, the paper lacks a theoretical framework in which researches on the topic of the present study are compared and presented.
Response: Though we did not add an illustrative theoretical model which is difficult to ascertain at this early age of evidence in this area, we believe that our introduction gives the reader an overall outlook on the problem at hand, the COVID-19 pandemic and its impact on women of reproductive age group and provide how we plan on reviewing published data for it. We have particularly presented and compared other researches in the ‘Discussion’ section (line no. 183; Page 18)
Comment 3: Another aspect to highlight is the absence of hypotheses or research questions, as well as the lack of mention of the research gap found and which has led them to carry out the study.
Response: The research question has been added to the first line of last paragraph in the ‘Introduction’ section (line no. 26-28; page 2)
Comment 4: I would like to highlight the good work done by the authors in the development of the methodology, allowing the reader to follow the process carried out for the selection of the articles.
I agree with the choice of the PRISMA model, but I also think it would be interesting to use another complementary technique such as the AMSTAR model to obtain more significant results in terms of the quality of the articles, as well as in the correct execution of an SLR. For its development, I would suggest that the authors read and cite the following article “ “Reyes-Menendez, A., Saura, J. R., & Filipe, F. (2019). The importance of behavioral data to identify online fake reviews for tourism businesses: A systematic review. PeerJ Computer Science, 5, e219” to reinforce the framework.
Similarly, I agree with the National Heart, Lung and Blood Institute (NHLBI) model used to verify the quality of observational studies.
Response: Thanks for the suggestion. We assessed the AMSTAR model and noticed that it is only for intervention studies. All the studies included in this review are descriptive studies only. Therefore, we cannot add the AMSTAR model to our manuscript.
Comment 5: In terms of results, the sample size chosen is correct for a systematic review of the literature. The process followed is clear and figure 1 helps for a visual understanding.
With regard to tables 1, 2a and 2b referring to the characteristics of each study and the methodology used, respectively, they are complex to understand, so I would suggest that the authors redraft them and develop the tables further. Regarding the table, I agree with its structure, but like the previous tables, it needs to be developed more deeply.
For a correct elaboration of the tables, I would suggest reading and citing the following article where the methodology is also a systematic review of the literature “Reyes-Menendez, A., Correia, M. B., Matos, N., & Adap, C. (2020). Understanding Online Consumer Behavior and eWOM Strategies for Sustainable Business Management in the Tourism Industry. Sustainability, 12(21), 8972”.
Response: Thanks for your suggestions, we have reviewed these again and though a bit lengthy, these tables provide important information which cannot be cut down. We can divide Table 1 into two separate tables, one with the study characteristics (study ID, study design, country and time period, setting and total number) and the other table with study details on pregnant and non-pregnant women (demographics, past medical history, presenting signs and symptoms, management and complications). However, this division will increase the number of tables in the manuscript and the journal has limitations to that.
Comment 6: With regard to the authors exposed in the Sensitive Analysis point: "The studies conducted by Zambrano et al. and Martinez-Portilla et al." (first line, first paragraph), the authors should follow the APA style for citing authors, where according to this format, the name of the author(s) plus the date of publication of the research must be stated.
Response: Thank you. We have edited the format in the manuscript as per this comment to name of the author and year of publication (Zambrano 2020 and Martinez-Portilla 2021).
Comment 7: Correct development of the discussion making reference to most of the points in the paper, however, I would suggest the authors mention the methodology used, as well as compare their research with a larger number of authors than those presented to reinforce it.
Response: We have now mentioned that this was a systematic review. Due to limited research/systematic reviews conducted on this topic, we did not have a lot of articles we could compare our results with.
Comment 8: I agree both with the six limitations raised and with the possible lines of future work. To reinforce this discussion point, I would offer the authors to outline the possible practical implications of their research, which, as I mentioned at the beginning of the review, could be endless for health professionals, researchers, scientists and others.
Response: We have stated the importance of this review in the last paragraph of ‘Introduction’ (line no. 24; Page 2) section and in the ‘Conclusion’ section (line no. 250; Page 19).
Comment 9: The conclusion is very brief, it should be developed with a greater magnitude making reference to all the points of the study.
Response: The significant results and points of the study have been mentioned in the second paragraph of ‘Discussion’ section and reporting it in conclusion would seem like repetition. This would also increase word count which we do not have room to increase further.
Comment 10: With regard to the references, for a scientific article there is a small number of bibliographical citations and they are not in accordance with the required APA style. However, they are up to date.
Response: We have referenced the review as per the ACS style, which is the requirement of this journal.
Comment 11: In conclusion, I find the research very interesting and with our suggestions, it can be considered for publication.
Response: Thanks for the appreciation.
Round 2
Reviewer 1 Report
The authors have responded to my comments adequately. The manuscript is acceptable for publication at the present form.